# Machine-Learning Algorithm for Predicting Fatty Liver Disease in a Taiwanese Population

**DOI:** 10.3390/jpm12071026

**Published:** 2022-06-23

**Authors:** Yang-Yuan Chen, Chun-Yu Lin, Hsu-Heng Yen, Pei-Yuan Su, Ya-Huei Zeng, Siou-Ping Huang, I-Ling Liu

**Affiliations:** 1Department of Internal Medicine, Division of Gastroenterology, Changhua Christian Hospital, Changhua 500, Taiwan; 27716@cch.org.tw (Y.-Y.C.); 111252@cch.org.tw (P.-Y.S.); 120693@cch.org.tw (Y.-H.Z.); 182972@cch.org.tw (S.-P.H.); 125267@cch.org.tw (I.-L.L.); 2Department of Hospitality Management, MingDao University, Changhua 500, Taiwan; 3Department of Family Medicine, Asia University Hospital, Taichung 400, Taiwan; amonslin@gmail.com; 4Chun-Shen Clinic, Nantou 540, Taiwan; 5General Education Center, Chienkuo Technology University, Changhua 500, Taiwan; 6Department of Electrical Engineering, Chung Yuan Christian University, Taoyuan 320, Taiwan; 7Artificial Intelligence Development Center, Changhua Christian Hospital, Changhua 500, Taiwan; 8College of Medicine, National Chung Hsing University, Taichung 400, Taiwan

**Keywords:** machine learning, fatty liver disease, predicting

## Abstract

The rising incidence of fatty liver disease (FLD) poses a health challenge, and is expected to be the leading global cause of liver-related morbidity and mortality in the near future. Early case identification is crucial for disease intervention. A retrospective cross-sectional study was performed on 31,930 Taiwanese subjects (25,544 training and 6386 testing sets) who had received health check-ups and abdominal ultrasounds in Changhua Christian Hospital from January 2009 to January 2019. Clinical and laboratory factors were included for analysis by different machine-learning algorithms. In addition, the performance of the machine-learning algorithms was compared with that of the fatty liver index (FLI). Totally, 6658/25,544 (26.1%) and 1647/6386 (25.8%) subjects had moderate-to-severe liver disease in the training and testing sets, respectively. Five machine-learning models were examined and demonstrated exemplary performance in predicting FLD. Among these models, the xgBoost model revealed the highest area under the receiver operating characteristic (AUROC) (0.882), accuracy (0.833), F1 score (0.829), sensitivity (0.833), and specificity (0.683) compared with those of neural network, logistic regression, random forest, and support vector machine-learning models. The xgBoost, neural network, and logistic regression models had a significantly higher AUROC than that of FLI. Body mass index was the most important feature to predict FLD according to the feature ranking scores. The xgBoost model had the best overall prediction ability for diagnosing FLD in our study. Machine-learning algorithms provide considerable benefits for screening candidates with FLD.

## 1. Introduction

Nonalcoholic fatty liver disease (NAFLD) is a hepatic complication of metabolic syndrome. With the elimination of hepatitis C and effective vaccination against hepatitis B, NAFLD is becoming the most common chronic liver disease in the world, affecting 22.28%–51.04% of the Asian population [1]. Identifying potential patients at increased risk of developing NAFLD [2] is important for early medical interventions to reduce their subsequent risk of developing liver cirrhosis and hepatocellular carcinoma [1]. The golden standard diagnosis of NAFLD is liver biopsy, which is invasive and not applicable for screening purposes. Noninvasive imaging methods, such as the controlled attenuation parameter (CAP) measurement with the FibroScan [3,4,5,6] or abdominal ultrasound, are able to screen patients for the presence of NAFLD and stage its severity. However, the routine use of these imaging tools may not be cost-effective and not widely available for primary care physicians. An ideal tool should allow for screening to be performed on a day-to-day basis, and, more importantly, increase the acceptance of screening by the physicians performing it. The fatty liver index (FLI) [7,8] is developed and validated with routine laboratory values for screening purposes when an ultrasound is not available. The FLI includes four parameters but is not easily calculated with simple calculation. Machine learning (ML) has recently been introduced to manage a large amount of data with the use of a computer that studies interactions between variables through the minimization of errors between the predicted and actual outcomes. Several ML techniques, such as logistic regression (LR), random forest (RF), artificial neural networks (ANNs), support vector machines, and extreme gradient boosting (xgBoost), show promise in improving predictions compared with conventional risk scoring systems. There are several previous studies that used ML methods to show a higher diagnostic value for the presence of fatty liver disease with clinical variables [9,10,11,12,13,14,15]. However, these studies utilized a limited number of datasets, and most of them did not examine with an additional testing dataset for validation. In this study, we utilized a health checkup database with a large Taiwanese population to evaluate the potential usefulness of different types of machine-learning algorithms. The performance of the machine-learning algorithms was compared with that of the well-known fatty liver index (FLI).

## 2. Materials and Methods

### 2.1. Patients and Data Preparation

The study population was recruited from adults (≥20 years old) who had received health examinations in the Healthcare Center of Changhua Christian Hospital between Jan 2009 and Jan 2019. The enrollment was limited to participants who had complete records of clinical and biochemical data, and results of liver ultrasonography with report for the presence of fatty liver disease. Patients with an ultrasound finding of hepatic malignancy, liver cirrhosis, ascites, or features of alcoholic liver disease were excluded. A total of 31,930 subjects who fulfilled the criteria were included in the study.

This study only accessed deidentified data retrospectively, so we waived the requirement for informed consent; the study was approved by the Institutional Review Board of Changhua Christian Hospital (approval number: CCH IRB 191012).

### 2.2. Diagnosis of Fatty Liver Disease (FLD)

The diagnosis of fatty liver disease requires the presence of significant hepatic steatosis confirmed by ultrasound examination. Three experienced sonographers who were unaware of the patients’ clinical and laboratory data performed the hepatic ultrasonography examinations during the study period. An ultrasound finding of moderate-to-severe fatty liver was defined as the presence of fatty liver disease.

### 2.3. Machine-Learning Model Construction and Validation

The dataset was randomly divided into a training set and a testing dataset at a ratio of 8:2. The training was performed with 10-fold cross-validation of the training data. The performance of the developed model was evaluated on the testing dataset. All clinical and biochemical data from the participants were used to build five models to predict the presence of FLD: extreme gradient boosting (xgBoost), logistic regression (LR), neural network (NN), random forest (RF), and support vector machine (SVM) models.

### 2.4. Performance Metrics

The six evaluation indicators of the area under the receiver operating characteristic curve (AUROC), accuracy, recall, F1 score, precision, and specificity were evaluated to compare the performance of the five models [16]. The AUROC is a popular and strong metric in evaluating binary classifiers. The AUROCs of the fatty liver index [8] and the developed machine models were compared.

### 2.5. Statistical Analysis

In our experiment, the machine-learning training testing was performed with the Orange Data Mining platform [17]. Baseline data were analyzed using IBM SPSS version 28.0 (IBM Corp., Armonk, NY, USA) and medical statistical software MedCalc Version 19.8 (© 2022 MedCalc Software Ltd.). Results were considered to be statistically significant if the two-tailed *p*-value was <0.05 for all tests.

## 3. Results

### 3.1. Characteristics of the Participant Population

A total of 31,930 subjects who fulfilled the criteria were included in the study, and 26% of the study population had moderate-to-severe fatty liver disease on abdominal ultrasound examination. Table 1 illustrates the clinical features of the fatty-and nonfatty-liver populations. A total of 27 features were obtained for the database, and all had statistical differences in these two populations. The dataset was split into a training dataset and a testing dataset (Table 2) for machine-learning model training.

### 3.2. Results of Different Model Performance Metrics

As there was a difference in the features of the fatty and nonfatty groups, we included all these variables in building the final model. The training of the machine-learning model was performed with Orange software (Version 3.31.1).

Table 3 illustrates the performance metrics of five different machine models. The xgBoost model had the highest AUROC for predicting the presence of fatty liver disease compared with that of the four other models. The SVM model had the worst performance metrics in predicting the presence of fatty liver disease. Figure 1 illustrates the top ten features contributing to the F1 score of the developed xgBoost model.

### 3.3. Comparison of the Performance of Machine-Learning Models and the Fatty Liver Index

The fatty liver index (FLI) is a conventional index developed to calculate the likelihood of fatty liver disease utilizing four clinical parameters: BMI, waist, serum triglyceride, and serum gamma-glutamyl transpeptidase (rGT) levels. FLI = (e0.953 × loge (triglycerides) + 0.139 × BMI + 0.718 × loge (rGT) + 0.053 × waist circumference − 15.745)/(1 + e0.953 × loge (triglycerides) + 0.139 × BMI + 0.718 × loge (rGT) + 0.053 × waist circumference − 15.745) × 100. A comparison of the developed machine-learning models and the FLI is illustrated in Table 4.

FLI had a statistically lower AUROC than those of the xgBoost and logistic regression models in the testing dataset, but a higher AUROC than that of the SVM model (Figure 2). Figure 3 illustrates the comparison of the precision–recall curve of the xgBoost model and fatty liver index. The xgBoost model had a larger AUC in the precision–recall curve than that of the fatty liver index.

The area under the curve of the fatty liver index was 0.648 (0.624 to 0.671), and that of the xgBoost model was 0.723 (0.701 to 0.744).

## 4. Discussion

Our study compared the performance of machine-learning models and the fatty liver index for the diagnosis of fatty liver disease in a hospital setting of a Taiwanese population. Fatty liver disease was observed in 26.2% of our patients. To our knowledge, only a few studies reported the use of machine learning for the diagnosis of fatty liver disease, and our study utilized a large dataset for model training and testing. The machine-learning algorithms achieved better performance than that of the conventional fatty liver index.

After hepatitis B vaccination and hepatitis C elimination [18,19,20], fatty liver disease has become the most health-threatening liver disease in the world [1,21,22]. The prevalence of fatty liver disease has seen a rapid rise in the Asian population, with the highest prevalence in Iran (64.29%) and the lowest in Taiwan (30.79%) [1]. Thus, identifying patients at risk for harboring fatty liver disease is important for subsequent lifestyle intervention to prevent liver damage progression. An abdominal ultrasound is radiation-free and noninvasive in screening the liver for the presence of fatty liver disease. A good sensitivity (85%–96%) and specificity of up to 98% could be achieved when moderate-to-severe fatty liver is detected by the abdominal ultrasound [14]. Despite its usefulness, an abdominal ultrasound cannot be available for every primary care setting or for population-based screening. In the 2022 clinical practice guideline by the American Association of Clinical Endocrinology and American Association for the Study of Liver Diseases [23], screening high-risk patients (prediabetes, type 2 diabetes, obesity, and/or metabolic syndromes) with noninvasive biomarkers such as the fatty liver index is recommended, followed by referral ultrasound examination. Such two-step screening is feasible, and may reduce screening time and costs [24]. A machine-learning model could be built into the hospital electronic system or in the form of internet apps, which may further decrease the difficulty for clinical use.

Recent advances in the field of machine learning have improved the discovery of new biomarkers for disease diagnosis or helped in designing treatment plans [14,25,26,27]. Dundar et al. [28] utilized a proposed machine-learning surgical planning and found that it significantly contributed to positive outcomes for neurosurgery. Sakatani et al. [29] utilized a machine-learning approach to estimate human cerebral atrophy on the basis of metabolic status. Shiba et al. [30] identified high risk factors for COVID-19 infection and hospitalization utilizing UK biobank data with machine-learning-based analysis. In addition, there are several previous works applying machine-learning methods for diagnosing NAFLD by utilizing electronic medical records or biochemical variables (Table 5). Even in the same ethnic population, the proportion of fatty liver disease could range from the 26.2% in our study to the 65.3% in a previously published study [15]. The rank of feature importance can, therefore, be different among different studies. Thus, the high accuracy of the developed model [15] may not be applicable to other populations. Different studies utilized different features for model training, and their performance could not be compared head-to-head. The machine-learning models should be compared with other validated tools such as the fatty liver index in the present study to show their superiority.

Our study evaluated five commonly utilized machine-learning models on the basis of available clinical biochemical variables in a health checkup setting. We compared the predictive capability of seven advanced machine-learning methods and confirmed that the xgBoost model demonstrated the best performance, with the highest AUROC (0.882). High accuracy was found in previous studies utilizing machine-learning methods [14,15,31,32], and the xgBoost model achieved better performance in our study. The xgBoost model has many advantages over other machine-learning models. For example, xgBoost performs a second-order Taylor expansion on the cost function for more accurate results. This model adds a regular term to the cost function to control the complexity of the model, simplifying it and preventing overfitting with improved training speed. In addition, xgBoost is a model based on the decision tree model, and it is more explanatory than neural networks and other algorithms are [33].

Our study has the strength of having had a large dataset for model training and testing as compared to those in previous reports. The use of big data may avoid overfitting the trained model. Despite the utilization of a large dataset for model training in a previous study [33], there was no comparison of the model’s performance with that of an existing validated index as in the present study. The use of the developed model may require validation before its application in clinical practice. In addition, our study included laboratory markers there are readily available in health check-ups in the majority of hospitals in Taiwan. Thus, our developed model may be more practicable in clinical practice.

Our findings were consistent with the findings of Atsawarungruangkit et al. [13], who demonstrated the superiority of a machine-learning model over the fatty liver index in predicting the presence of fatty liver disease, although the machine model utilized more features than the fatty liver index did and could not be calculated with a calculator. The calculation of the fatty liver index was also not simple and required the use of a spreadsheet or an internet app that would be similar to the use of a machine model. Utilizing a machine-learning model with better performance could assist in effectively identifying fatty liver disease in future clinical practice.

There are several limitations to the present study. First, we did not incorporate the clinical information of the patients that was not included in our database. The presence of diabetes mellitus, hepatitis B, hepatitis C, and medication used may influence the findings of the ML models in predicting the presence of fatty liver disease. Further studies are required to include this information to improve the models’ performance. Second, the database did not include a history of alcohol consumption. Although we could exclude the presence of significant liver disease or alcoholic liver disease, we may have included a small proportion of patients with alcoholic fatty liver disease in our analysis. Thus, the final prediction of fatty liver disease may not be valid for other patient or ethnic populations. Third, our laboratory values did not include the level of uric acid, which is not a routine examination in our cohort for health check-ups. As uric acid level was identified as a potential marker for predicting fatty liver disease [32], the lack of this parameter may have influenced the predicting ability of our ML models. In addition, as a health check-up cohort, there were no biopsy data to confirm the extent of steatosis and the severity of liver fibrosis. Further studies including these parameters as features may further improve such ML models by providing more information on the likelihood of disease severity, and predicting patient mortality risk, and the extent of steatosis and fibrosis.

## 5. Conclusions

The present study utilized a large dataset, and the xgBoost model had the best overall prediction ability for diagnosing FLD in our population. Furthermore, machine-learning algorithms provided considerable benefits for screening candidates with FLD.

## Figures and Tables

**Figure 1 jpm-12-01026-f001:**
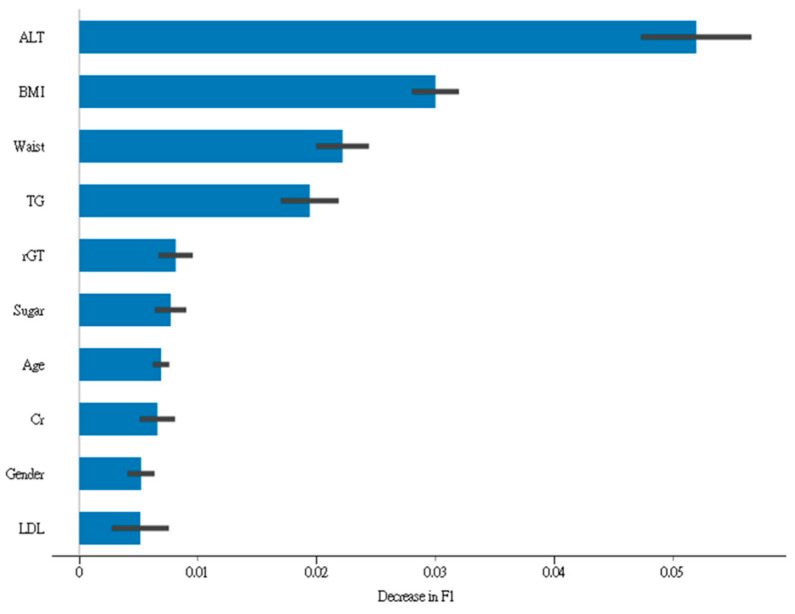
Top ten features of data contributing to the F1 score of the developed xgBoost model.

**Figure 2 jpm-12-01026-f002:**
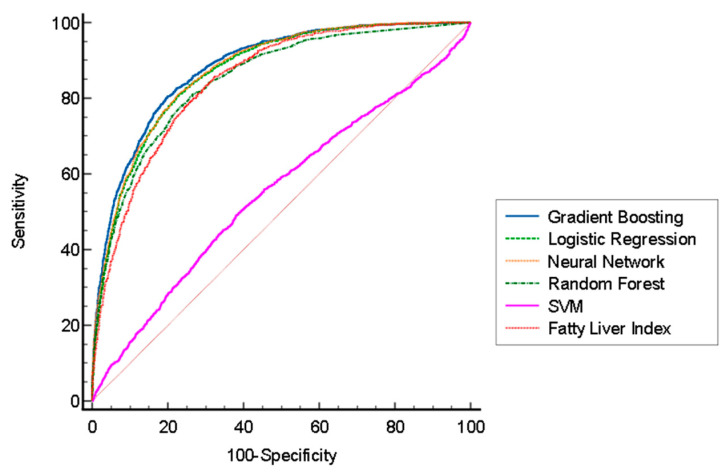
AOC curve of five different machine-learning models and the fatty liver index.

**Figure 3 jpm-12-01026-f003:**
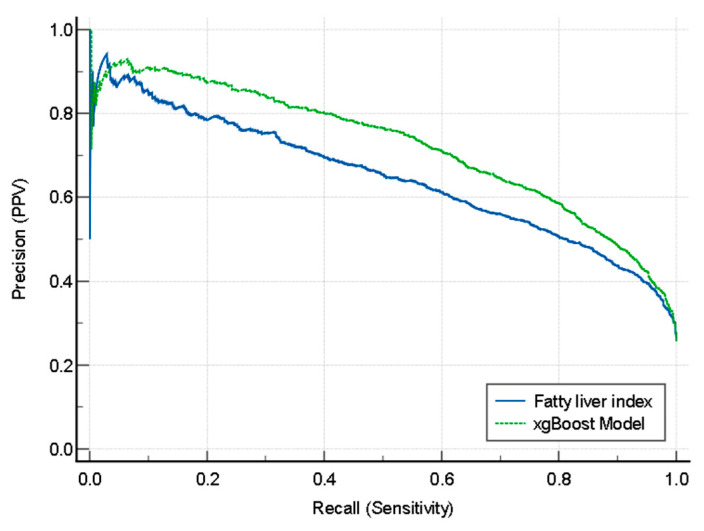
Comparison of the precision–recall curve of the xgBoost model and fatty liver index.

**Table 1 jpm-12-01026-t001:** Comparison of the fatty and nonfatty populations.

	No Fatty Liver (*n* = 23,625)	Fatty Liver Disease (*n* = 8305)	*p*-Value
Categorial variable	N (%)	N (%)	
Male sex	13,484 (57.1%)	6293 (75.8%)	<0.0001
Continuous variables	Mean ± SD	Mean ± SD	
Age (years)	48.63 ± 10.92	50.48 ± 9.93	<0.0001
Weight (kg)	63.28 ± 10.6	75.011 ± 12.13	<0.0001
Height (cm)	164.7 ± 8.02	166.34 ± 7.95	<0.0001
BMI (kg/m^2^)	23.244 ± 2.91	27.044 ± 3.47	<0.0001
Waist (cm)	63.27 ± 10.6	75.01 ± 12.13	<0.0001
SBP (mmHg)	121.11 ± 16.14	130.17 ± 15.77	<0.0001
DBP (mmHg)	77.25 ± 10.42	83.45 ± 10.62	<0.0001
ALT (IU/L)	23.31 ± 20.42	39.64 ± 25.71	<0.0001
AST (IU/L)	23.86 ± 19.344	30.39 ± 16.33	<0.0001
Cr (mg/dL)	0.811 ± 0.23	0.86 ± 0.23	<0.0001
Sugar (mg/dL)	93.88 ± 16.01	104.44 ± 25.56	<0.0001
T-Cho (mg/dL)	191.666 ± 34.5	197.31 ± 36.39	<0.0001
HDL (mg/dL)	52.588 ± 13.53	43.07 ± 9.39	<0.0001
LDL (mg/dL)	118.4 ± 30.38	124.08 ± 32.47	<0.0001
TG (mg/dL)	98.52 ± 69.62	162.64 ± 110.33	<0.0001
r-GT (U/L)	21.91 ± 24.66	35.08 ± 36.76	<0.0001
WBC (×10^9^/L)	5.4 ± 1.45	6.18 ± 1.56	<0.0001
Hb (g/dL)	13.99 ± 1.53	14.7 ± 1.31	<0.0001
MCH (pg)	30.11 ± 2.98	30.17 ± 2.71	0.1025
MCHC (g/dL)	33.455 ± 0.95	33.61 ± 0.94	<0.0001
MCV (fL)	41.8 ± 4.21	43.71 ± 3.63	<0.0001
RBC-RDW (%)	13.53 ± 1.27	13.39 ± 1.04	<0.0001
RBC Count (10^6^/μL)	4.67 ± 0.52	4.9 ± 0.51	<0.0001
RBC Volume (fL)	89.89 ± 7.53	89.67 ± 6.81	0.0166
Platelet (10^3^/μL)	222.64 ± 53.55	229.82 ± 52.68	<0.0001
FIB-4	1.21 ± 0.64	1.17 ± 0.56	<0.0001

Abbreviations: SBP: systolic blood pressure; DBP: diastolic blood pressure; ALT: alanine aminotransferase; AST: aspartate aminotransferase; Cr: creatinine; T-Cho: total cholesterol; HDL: high-density lipoprotein; LDL: low-density lipoprotein; TG: triglyceride; r-GT: r-glutamyl transpeptidase; WBC: white blood cell count; HB: hemoglobin; MCH: mean corpuscular hemoglobin; MCHC: mean corpuscular hemoglobin concentration; MCV: mean corpuscular volume; RBC: red blood cell; RDW: red cell distribution width; FIB-4: fibrosis index based on the four factors.

**Table 2 jpm-12-01026-t002:** Baseline data of the testing and training population**.**

	Testing Population (*n* = 6386)		Training Population (*n* = 25,544)		*p*-Value
Categorial Variable	N	%	N	%	
Male sex	3920	61.4%	15857	62.1%	0.3077
Fatty liver disease	1647	25.8%	6658	26.1%	0.6552
Continuous variables	Mean	SD	Mean	SD	
Age (years)	49.0338	10.6897	49.1273	10.7045	0.5325
Weight (kg)	66.2086	11.8944	66.3478	12.2305	0.4133
Height (cm)	165.1039	8.0304	165.1287	8.0323	0.8254
BMI (kg/m²)	24.1912	3.3860	24.2331	3.5129	0.3896
Waist (cm)	81.3276	9.4895	81.4931	9.6282	0.2178
SBP (mmHg)	123.3472	16.2762	123.4969	16.5927	0.5173
DBP (mmHg)	78.7839	10.8325	78.8869	10.8106	0.4962
ALT (IU/L)	27.6682	20.3123	27.5262	23.6926	0.6599
AST (IU/L)	25.4887	11.4994	25.5720	20.2435	0.7520
Cr (mg/dL)	0.8184	0.2498	0.8216	0.2228	0.3200
Sugar (mg/dL)	96.8274	21.5711	96.5744	18.9770	0.3543
T-Cho (mg/dL)	192.5857	34.7903	193.2651	35.1615	0.1663
HDL(mg/dL)	50.2668	13.2300	50.0619	13.2686	0.2693
LDL (mg/dL)	119.4998	30.8285	119.9765	31.0895	0.2723
TG (mg/dL)	113.8447	101.3611	115.5403	82.8250	0.1629
r-GT (U/L)	25.2388	28.2466	25.3584	29.0509	0.7674
WBC (×10^9^/L)	5.6059	1.5110	5.6049	1.5196	0.9632
Hb (g/dL)	14.1648	1.5050	14.1758	1.5102	0.6019
MCH (pg)	30.1294	2.8629	30.1250	2.9260	0.9135
MCHC (g/dL)	33.4929	0.9346	33.4895	0.9532	0.7981
MCV (fL)	42.2640	4.1367	42.3027	4.1600	0.5058
RBC-RDW (%)	13.5007	1.2364	13.4978	1.2132	0.8666
RBC Count (10^6^/μL)	4.7246	0.5119	4.7314	0.5298	0.3560
RBC volume (fL)	89.8414	7.2438	89.8342	7.3755	0.9443
Platelet (10^3^/μL)	225.1690	52.8910	224.3424	53.5484	0.2688
FIB-4	1.1890	0.6050	1.1966	0.6242	0.3836

Abbreviations: SBP: systolic blood pressure; DBP: diastolic blood pressure; ALT: alanine aminotransferase; AST: aspartate aminotransferase; Cr: creatinine; T-Cho: total cholesterol; HDL: high-density lipoprotein; LDL: low-density lipoprotein; TG: triglyceride; r-GT: r-glutamyl transpeptidase; WBC: white blood cell count; HB: hemoglobin; MCH: mean corpuscular hemoglobin; MCHC: mean corpuscular hemoglobin concentration; MCV: mean corpuscular volume; RBC: red blood cell; RDW: red cell distribution width; FIB-4: fibrosis index based on the four factors.

**Table 3 jpm-12-01026-t003:** Performance of different machine models on the testing dataset.

Model	AUROC	Accuracy	Recall	F1	Specificity	Precision
xgBoost	0.882	0.833	0.833	0.829	0.683	0.827
Neural network	0.874	0.824	0.824	0.820	0.683	0.818
Logistic regression	0.870	0.825	0.825	0.815	0.629	0.816
Random forest	0.849	0.818	0.818	0.809	0.629	0.808
SVM	0.551	0.569	0.569	0.595	0.536	0.656

Abbreviations: AUROC: area under receiver operating characteristic curve; SVM: support vector machine.

**Table 4 jpm-12-01026-t004:** Pairwise comparison of the AUROC of different machine-learning models and the fatty liver index on the testing dataset.

Difference between Areas (*p*-Value)	Neural Network	Logistic Regression	Random Forest	SVM	Fatty Liver Index
xgBoost	0.0076 (*p* = 0.0105)	0.0114 (*p* = 0.0001)	0.0327 (*p* < 0.0001)	0.330 (*p* < 0.0001)	0.0347 (*p* < 0.0001)
Neural network		0.00382 (*p* = 0.2303)	0.0251 (*p* < 0.0001)	0.323 (*p* < 0.0001)	0.00204 (*p* = 0.5978)
Logistic regression			0.0213 (*p* < 0.0001)	0.0319 (*p* < 0.0001)	0.0233 (*p* < 0.0001)
Random forest				0.298 (*p* < 0.0001)	0.00204 (*p* = 0.5978)
SVM					0.295 (*p* < 0.0001)

Abbreviations: AUROC: area under receiver operating characteristic curve; SVM: support vector machine.

**Table 5 jpm-12-01026-t005:** Literature review of previous studies of machine learning for fatty liver disease.

Author/Year	Setting/Country	Fatty/Total Population, (%)	Validation Method	ML Model	Accuracy (%)	Area under Curve (%)
Ma [31] 2018	Hospital/China	2522/10,508 (24%)	10-fold cross validation	LR	82.92%	N/A
Wu [15] 2018	Hospital/Taiwan	377/577 (65.3%)	10-fold cross validation	Random forest	87.48%	92.25%
Liu [14] 2021	Hospital/China	5878/15,315 (38.4%)	32% of dataset as testing data	xgBoost	79.5%	87.3%
Atsawarungruangkit [13] 2021	Population/USA	817/3235 (25.3%)	30% of dataset as testing data	Ensemble of subspace discriminant	77.7%	78%
Pei [32] 2021	Hospital/China	845/3419 (24.7%)	30% of dataset as testing data	xgBoost	94.15%	93.06%
Zhao [33] 2021	Hospital/China	9173/39,884 (23%)	30% of dataset as testing data	xgBoost	89%	N/A
Our Study 2022	Hospital/Taiwan	8375/31,930 (26.2%)	20% of dataset as testing data	xgBoost	83.3%	88.2%

## Data Availability

Data are available on reasonable request.

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
