# Peer review of "Machine-Learning Algorithm for Predicting Fatty Liver Disease in a Taiwanese Population"

_jpm, 2022, doi:10.3390/jpm12071026_

Round 1
Reviewer 1 Report
1. Table 1~2: The decimal point is unified with 1~2 digits, the categorical variable is expressed as n(%), and the continuous variable is expressed as mean±SD
2. Describe the criteria for variable selection and modeling building in Materials and Methods in detail.
3. Materials and Methods: Describe the criteria for variable selection and modeling building in detail.
4. Table 3: A detailed description of the feature ranking methods is required. Information Gain, Gain Ratio, and Gini ratio are the impurity indexes of the decision tree. There is no detailed explanation of the research method.
5. Line 229: identify fatty disease -> identify fatty liver disease
6. Compare the papers below in the discussion.
(https://www.sciencedirect.com/science/article/pii/S0169260718315724)
Author Response
Dear Reviewer,
Thank you for reviewing our manuscript and providing your editorial comments as well as reviewers comments for improving our manuscript. Based on these comments, we have made several revisions to our manuscript, which we are hereby resubmitting for your consideration. Our point-by-point responses to the comments are detailed below.
Response to Reviewers’ comments
- Table 1~2: The decimal point is unified with 1~2 digits, the categorical variable is expressed as n(%), and the continuous variable is expressed as mean±SD
Response: Thank you for your comment. We made revisions according to your suggestion in the revised manuscript.
- Describe the criteria for variable selection and modeling building in Materials and Methods in detail.
- Materials and Methods: Describe the criteria for variable selection and modeling building in detail.
- Table 3: A detailed description of the feature ranking methods is required. Information Gain, Gain Ratio, and Gini ratio are the impurity indexes of the decision tree. There is no detailed explanation of the research method.
Response: Thank you for your comment. As the data came from health checkup data in our hospital. There is no missing data with our predefined inclusion criteria. As the features are significantly different in the FLD vs non-FLD group, we input all features in the model for training in the Orange Data Mining platform.
- Line 229: identify fatty disease -> identify fatty liver disease
Response: Thank you for your comment. We made revisions according to your suggestin.
- Compare the papers below in the discussion. (https://www.sciencedirect.com/science/article/pii/S0169260718315724)
Response: Thank you for your comment. This paper is cited and we add some discussion in the revised manuscript.
Thank you for the opportunity to resubmit this manuscript for consideration for publication in the Journal of Personalized Medicine If you have any questions or comments regarding this manuscript, please do not hesitate to contact us using the details provided below.
Sincerely,
Hsu-Heng Yen, M.D
Division of Gastroenterology, Department of Internal Medicine, Changhua Christian Hospital, Changhua, Taiwan
Fax: +886-4-7228289
Tel: +886-4-7238595ext5501
E-mail: 91646@cch.org.tw
Reviewer 2 Report
This is an interesting study from a large cohort of patients which addresses the important clinical challege of identifying prgnostic markers in patients with NAFLD. It is particularly important to identify markers that work for specific international populations as there are differences in performance of key non invasive test platforms between global patients populations with different extents of obesity etc.
The methods section could perhaps be more detailed to characterise exactly what parameters were used in each model and how the analysis was performed. This is important as otherwise its not easy to understand how some of the models have different predictive capacities and what the prediction is telling you (liklihood of severe disease, mortality risk, extent of steatosis/fibrosis?). It seems that outcome is compared to the FLI but it would be useful to see perfromance vs scores like APRI/NAFLD fibrosis score or FIB-4
The authors clearly describe the characteristics of their FLD vs no FLD cohorts and state that ultrasound had been performed to stratify the groups. However the ultrasound findings are not captured in the table. Given there is no biopsy data to confirm extent of steatosis and the FIB-4 scores are suggestive of advanced fibrosis (NASH) in the "FLD" group it is important to consider whether the machine learning approach is identifying severe disease rather than steatosis per se. Here the term 'fatty liver disease' needs careful use in the manuscript and the authors should explain disease stage clearly as they consider their findings.
In the discussion the authors should consider the utility of their approach vs FIB-4 which is widely used, and suggest whether it offers advantages over such scores.
Similarly the authors suggest that one of the advantages of their approach is that not all centres have access to ultrasound. Would they have access to this machine learning system though and why wouldnt they use other common non-invasive blood tests? The authors need to clarify the value and novelty of their approach in the discussion.
Author Response
Dear Reviewer,
Thank you for reviewing our manuscript and providing your editorial comments as well as reviewers comments for improving our manuscript. Based on these comments, we have made several revisions to our manuscript, which we are hereby resubmitting for your consideration. Our point-by-point responses to the comments are detailed below.
Response to Reviewers’ comments
This is an interesting study from a large cohort of patients which addresses the important clinical challege of identifying prgnostic markers in patients with NAFLD. It is particularly important to identify markers that work for specific international populations as there are differences in performance of key non invasive test platforms between global patients populations with different extents of obesity etc.
Response: Thank you for your comment.
The methods section could perhaps be more detailed to characterise exactly what parameters were used in each model and how the analysis was performed. This is important as otherwise its not easy to understand how some of the models have different predictive capacities and what the prediction is telling you (liklihood of severe disease, mortality risk, extent of steatosis/fibrosis?). It seems that outcome is compared to the FLI but it would be useful to see perfromance vs scores like APRI/NAFLD fibrosis score or FIB-4
Response: Thank you for your comment. As the comparison of different diagnostic tools requires a reference diagnostic standard. Our patients didn’t have a liver biopsy to estimate the degree of liver fibrosis. In the method section we described how we define the presence of fatty liver disease as follows: The diagnosis of fatty liver disease requires the presence of significant hepatic steatosis confirmed by the ultrasound examination. Three experienced sonographers performed the hepatic ultrasonography examinations during the study period who were unaware of the patient clinical and laboratory data. An ultrasound finding of moderate to severe fatty liver disease was defined as the presence of fatty liver disease in the present study. We total agree with us that further studies are required to build a more comprehensive model that could provide more information on the likelihood of disease severity, mortality risk, and extent of steatosis/fibrosis.
The authors clearly describe the characteristics of their FLD vs no FLD cohorts and state that ultrasound had been performed to stratify the groups. However the ultrasound findings are not captured in the table. Given there is no biopsy data to confirm extent of steatosis and the FIB-4 scores are suggestive of advanced fibrosis (NASH) in the "FLD" group it is important to consider whether the machine learning approach is identifying severe disease rather than steatosis per se. Here the term 'fatty liver disease' needs careful use in the manuscript and the authors should explain disease stage clearly as they consider their findings.
Response: Thank you for your comment. Our patients didn’t have a liver biopsy to estimate the degree of liver fibrosis. In the method section, we described how we define the presence of fatty liver disease. An ultrasound finding of moderate to severe fatty liver disease was defined as the presence of fatty liver disease in the present study. We totally agree with us that further studies are required to build a more comprehensive model that could provide more information on the likelihood of disease severity, mortality risk, and extent of steatosis/fibrosis.
In the discussion the authors should consider the utility of their approach vs FIB-4 which is widely used, and suggest whether it offers advantages over such scores.
Response: Thank you for your comment. As FIB-4 is derived from the blood tests, including GOT/GPT and Platelet count. In our hospital electronic system, the FIB-4 score is calculated to estimate the degree of liver fibrosis. The present study aimed to develop a machine learning model to predict the presence of fatty liver disease that could be helpful in the daily practice to identify the high-risk patient for further ultrasound screening. Thus, the performance of our machine learning model could only be compared with the fatty liver index rather than the FIB-4 score.
Similarly the authors suggest that one of the advantages of their approach is that not all centres have access to ultrasound. Would they have access to this machine learning system though and why wouldnt they use other common non-invasive blood tests? The authors need to clarify the value and novelty of their approach in the discussion.
Response: Thank you for your comment. We believe the ultrasound may not be readily available and a machine learning system could be built in the hospital electronic system that is ready for use. Therefore, every physician using the electronic system or patients can have easy access to the machine model. Application of a machine learning model could be built into the hospital electronic system or in the form of internet Apps which may further decrease the difficulty for clinical use.
Thank you for the opportunity to resubmit this manuscript for consideration for publication in the Journal of Personalized Medicine If you have any questions or comments regarding this manuscript, please do not hesitate to contact us using the details provided below.
Sincerely,
Hsu-Heng Yen, M.D
Division of Gastroenterology, Department of Internal Medicine, Changhua Christian Hospital, Changhua, Taiwan
Fax: +886-4-7228289
Tel: +886-4-7238595ext5501
E-mail: 91646@cch.org.tw
Round 2
Reviewer 1 Report
The authors have satisfactorily responded to all my questions and made the necessary changes to the manuscript.